# Fine mapping epitope on Glycoprotein-Gn from Severe Fever with Thrombocytopenia Syndrome Virus

**Abulimiti Moming[1], Shen Shi[2], Shu Shen[3], Jie Qiao[3], Xihong Yue[2], Bo Wang[3], Juntao Ding[1], Zhihong Hu[3], Fei Deng[3]\*, Yujiang Zhang[2]\*, Surong Sun[1]\***

**1** Xinjiang Key Laboratory of Biological Resources and Genetic Engineering, College of Life Science and Technology, Xinjiang University, Urumqi, China, **2** Center for Disease Control and Prevention of Xinjiang Uygur Autonomous Region, Urumqi, China, **3** State Key Laboratory of Virology, Wuhan Institute of Virology, Chinese Academy of Sciences, Wuhan, China

\* df@wh.iov.cn (FD); xjsyzhang@163.com (YZ); sr_sun2005@163.com (SS)

## Abstract

Severe Fever with Thrombocytopenia Syndrome Virus (SFTSV) was recently identified as a tick-borne pathogen that threat to human health. Since 2010, many countries including China, South Korea, and Japan have reported Human SFTS caused by SFTSV infection. The glycoprotein encoded by the SFTSV M gene is the major antigenic component on the viral surface, and responsible for the viral entry, which makes it an important viral antigen and a clinical diagnostic target. The present study aimed to map linear B cell epitopes (BCEs) on the N-terminal glycoprotein (Gn) from SFTSV strain WCH/97/HN/China/2011 using the modified biosynthetic peptide method. Five fine epitopes (E1, $^{196}$FSQSEFPD$^{203}$; E2, $^{232}$GHSHKII$^{238}$; E3, $^{256}$VCY-KEGTGPC$^{265}$; E4, $^{285}$FCKVAG$^{290}$, and E5, $^{316}$SYGGM$^{320}$) were identified using the rabbit antisera. Western blot analysis showed that all the five epitopes interacted with the positive serum of sheep that had been naturally infected with SFTSV. Three-dimensional structural modeling analysis showed that all identified BCEs were located on the surface of the SFTSV-Gn and contained flexible loops. The sequence alignment revealed high conservation of the identified BCEs among 13 SFTSV strains from different lineage. These mapped epitopes will escalate the understanding of the epitope distribution and pathogenic mechanism of SFTSV, and could provide a basis for the development of a SFTSV multi-epitope detection antigen.

## Introduction

Severe Fever with Thrombocytopenia Syndrome Virus (SFTSV) is a human pathogen that causes Severe Fever with Thrombocytopenia Syndrome (SFTS), an emerging disease with high mortality rates up to 30% [1–3]. Since 2010, SFTSV is broadly disseminated in countries like China, South Korea, and Japan [4–6]. The clinical symptoms of SFTSV infections include fever, thrombocytopenia, gastrointestinal disorder, and leukocytopenia [7]. SFTSV is a tick-borne virus [8] and has infected many domestic animals, including goats, cattle, dogs, cats and chickens [9–12]. Humans could generally be infected through tick bites, direct contact with

**Data Availability Statement:** All relevant data are within the paper and its Supporting Information files

**Funding:** This work was supported partly by grants from the National Natural Science Foundation of

China (No. 81760365, 81690369 to S. R. S.), the Science Research Key Project of Xinjiang Education Department (No. XJEDU2019I002 to S. R. S.), and the Science and Technology Basic Work Program (No. 2013FY113500 to Y. J. Z) from the Ministry of Science and Technology of China, and funded by the Open Research Fund Program of the State Key Laboratory of Virology of China (No. 2015IOV003 to F. D.). The funders had no role in study design, data collection and analysis, decision to publish, or preparation of the manuscript.

**Competing interests:** The authors have declared that no competing interests exist.

blood or mucus of infected livestock, or patients [13]. Thus, early diagnosis and vaccine development are critical for the prevention and control of SFTSV.

SFTSV is classified as *Bandavirus* genus within the family *Phenuiviridae* [14]. The genome consists of three negative stranded RNAs, designated as large (L), medium (M), and small (S), which encode RNA polymerase, glycoprotein (GP) and nucleocapsid protein (NP), respectively. Studies have indicated that GP mediates the first step in the virus replication cycle of binding and entry into the host cell, and are the primary targets for neutralizing antibodies [15, 16]. The GP of *Phenuiviridae* are synthesized as precursor polyprotein in the infected cells, which can be cleaved by cellular proteases during translation, and processed into the mature virion subunits N-terminal glycoprotein (Gn) and C-terminal glycoprotein (Gc) [17], this step was executed by signal peptidase [18]. Gn/Gc is responsible for cell attachment and membrane fusion, which is required for host cell entry [15]. In the endoplasmic reticulum Gn and Gc are decorated with N-linked glycans, mature Gn and Gc are targeted to the Golgi apparatus for virus budding [19]. Gn and Gc are two major antigenic components on the viral surface, and are incorporated into the envelope of the virus particles. Finally, infected cells released infectious virus particles by exocytosis [17, 20]. Among them, the Gn protein plays a critical role in virion structure formation and adhesion to new target cells, rendering it the main target of neutralizing antibodies [16]. Therefore, elucidating the epitopes or immunodominant regions of Gn is critical for the development of vaccine design and diagnosis of viral infection.

Monoclonal antibodies (mAbs) or convalescent sera from SFTS patients were tested to identify potential therapeutic intervention targets, resulting in the identification of SFTSV-GP as molecules required for host cell entry and also as critical targets for virus neutralization through the development of humoral immunity [21, 22]. Specific treatment with antiviral agents targeting SFTSV is urgently needed to reduce the morbidity and mortality as much as possible. The mAb 10 [21] and mAb 4–5 [23] from SFTS patients targeting SFTSV-Gn were showing successful neutralizing activity to a variety of strains of SFTSV isolates in China [22]. Fine epitope motif mapping is an active field and one of the mapping approaches is called the biosynthetic peptides method (BSP). Overcome many obstacles in other mapping approaches, it has been recognized with many merits including simple design, outstanding cost-efficiency, high reliability, and adaptability to screening the entire region of interest [24]. Some research implies the BSP can be utilized for epitope motif identification [25, 26]. However, there have only been a few reports on epitope identification of SFTSV-Gn.

In this study, to identify immunodominant linear B cell epitopes (BCE) in the Gn (amino acid (aa) 189–451) of SFTSV, named as SGn, the modified BSP method was used to identify the fine epitopes of Gn from the SFTSV strain WCH/97/HN/China/2011 using rabbit polyclonal antibody (pAb) against SFTSV-Gn (α-SGn). We have identified five epitopes on the SGn by using the BSP method. All the five epitopes identified could be recognized by the antisera of sheep infected with SFTSV. We also analyzed the conservation of each epitope among homologous SGn proteins and their location in the predicted three-dimensional (3D) structure. All the five epitopes were distributed on the surface of SGn and thus facilitates antigen-antibody binding accessibility. These investigations have provided novel and comprehensive data about linear BCEs on SFTSV-Gn as well as their unique distribution profile, supported a more solid foundation for the design of SFTSV multi-epitope peptide diagnostics.

## Materials and methods

### Ethics statement

All methods were carried out in accordance with relevant guidelines and regulations. All experimental protocols were approved by the Committee on the Ethics of Animal Experiments

of Xinjiang Key Laboratory of Biological Resources and Genetic Engineering (Approval number: BRGE-AE001), Xinjiang University, and by the Ethics committees of Animal Experiments of Wuhan Institute of Virology, Chinese Academy of Sciences (Approval number: WIVH33201801). This study did not involve suffering or killing the animals.

## Plasmids and antibodies

The prokaryotic expression plasmid pMAL-c2x, and SFTSV strain WCH/97/HN/China/2011 were donated by Professor Fei Deng from Wuhan Institute of Virology, Chinese Academy of Sciences. The coding region (aa, 189–451) of SFTSV-Gn from strain WCH/97/HN/China/2011 was amplified by PCR using 2×Rapid Taq Master Mix (Vazyme Biotech, Nanjing, China) according to the manufacture's instruction. The PCR products were cloned into the plasmid pET-28a to generate the expression plasmid pET-28a-*SGn* and the insert was confirmed by sequencing. Protein expression and purification were conducted as described [27]. New Zealand rabbits were injected intramuscularly with 0.5 mg of purified SGn segment and immunized at two-week intervals according to the conventional animal immune method. After the third immunization for two weeks, rabbit antiserum was separated and stored at -80°C until use. The sheep serum samples used in the study were kindly provided by Professor Yujiang Zhang from Xinjiang Centers for Disease Control and Prevention (XJCDC). The SFTSV positive or negative sheep serum samples were previously identified using an immunofluorescence assay (IFA) and enzyme-linked immunosorbent assay (ELISA) [27]. SFTSV negative sheep serum and non-immunized New Zealand rabbit serum were used as negative controls in the Western blot and IFA, respectively. *Escherichia coli* BL21 (TB1 strain) competent cells were used to express 16/8/10mer peptides fused with a truncated MBP protein. Goat anti-rabbit and mouse anti-goat IgG conjugated to horseradish peroxidase (HRP) were purchased from Beijing TransGen Biotech, Co., Ltd. (Beijing, China).

## Other reagents and materials

DNA ligase and restriction enzymes *Bam*H I and *Sal* I (Takara Co., Ltd, Dalian, China), *E. coli* strain BL21 (TB1) competent cells (Novagen, Inc., Madison, USA), QIA quick Gel Extraction Kit (QIAGEN, Duesseldorf, Germany), 0.2 μm nitrocellulose membrane (Whatman GmbH, Dossel, Germany), unstained or prestained molecular weight markers (ThermoFisher Science, Waltham, MA, USA), and enhanced chemiluminescence (ECL) plus Western blot detection kit (GE Healthcare, Buckinghamshire, UK) were obtained. Other general chemicals were obtained from Shanghai Sangon Co., Ltd (Shanghai, China).

## Immunofluorescence Assay (IFA)

Vero cells were infected with SFTSV strain WCH/97/HN/China/2011 at an MOI of 5. At 72 h. p. i., the cells were fixed for 10 min with 4% paraformaldehyde-PBS. Then the fixed cells were incubated in 0.2% Triton X-100-PBS for permeabilization, and then blocked with 5% bovine-serum albumin (BSA). The cells were treated with rabbit pAb α-SGn (1:1000 dilution) as a primary antibodies for 2 h at room temperature or 4°C overnight, and incubated with goat anti-rabbit IgG-fluorescein isothiocyanate (FITC) (1:2000 dilution, TransGen Biotech, Beijing, China) for 1 h at room temperature. For visualization of the nuclei, the cells were incubated with Hoechst 33,258 (Beyotime, Shanghai, China) for 5 min at room temperature. All images were acquired using IX73 microscope (Olympus, Tokyo, Japan).

## Expression of truncated SGn fragment

In order to map the epitope region of SGn, series of trunctated fragments of SGn were cloned for expression (Fig 1). Primers were synthesized according to the sequence of strain WCH/97/HN/China/2011 (GenBank accession no. JQ341189.1). PCR was performed to amplify the fragments from plasmid pET-28a-*SGn* which contains the Gn aa 189–461 fragment of SFTSV. The clones were sequenced for verifying the veracity of the fragments. Then, the fragments were sub-cloned into prokaryotic expression vector pET-32a (+) with the *Bam*H I and *Hind* III sites on the 5'and 3'terminus. The expression vectors pET-32a-*SGn*1, pET-32a-*SGn*2, pET-32a-*SGn*3, pET-32a-*SGn*4 and pET-32a-*SGn*5 were identified by DNA sequencing (Sangon biotech, Shanghai). The fusion proteins were expressed in *E. coli* and SDS-PAGE was performed for detecting fusion protein expressions.

## Mapping strategy and biosynthesis of overlapping 16/8/10mer peptides

To map epitopes on the SGn1 and SGn2 segments, we used the feasible strategy shown in Fig 1. A total of 16 overlapping 16mer peptides spanning Gn numbered P1-P16 were bio-expressed (S1 Table). The 16mers all had an overlap of 8 aa residues between each two adjacent peptides. For fine epitopes motif mapping, eight sets of a total of 42 8mer peptides (named P17-P58) with an overlap of 7 aa residues and 6 10mer peptides (named P59-P64) with an overlap of 9 aa residues were bio-expressed based on the reactive 16mer peptides mapped in the first round of antigenic peptide mapping (S2 Table).

## Construction of recombinant plasmids

All plus and minus strands of DNA fragments that encoded target 16/8/10mer peptides and had cohesive end nucleotides of *Bam*H I and TAA-*Sal* I sites at the 5'- and 3'- ends were synthesized by Tianyi biotech Co., Ltd (Wuhan, China). Each plasmid expressing a 16/8/10mer

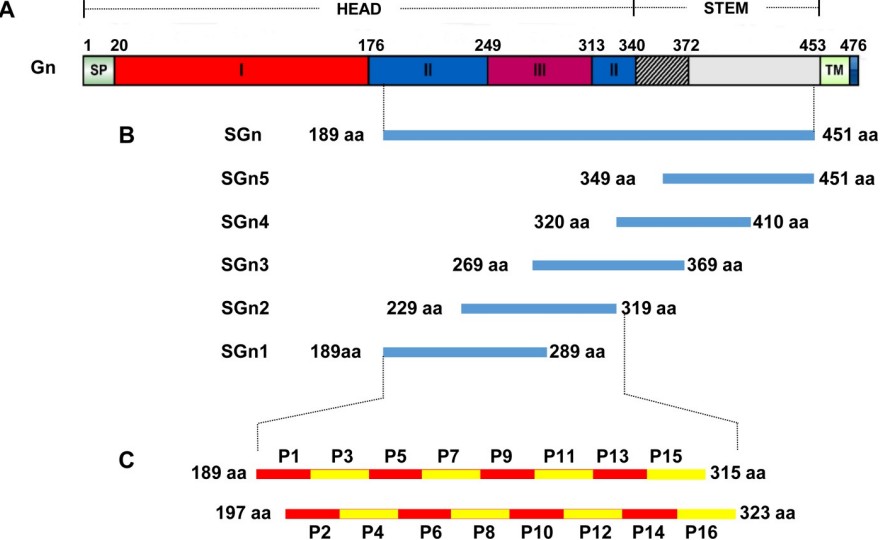

**Fig 1. Schematic representation of epitope mapping strategy.** (A) Schematic representation of the full length SFTSV-Gn [23], including Signal peptide (SP), three domains (I, red; II, blue; and III, purple), and Transmembrane region (TM). (B) Schematic representation of truncated fragments of Gn. The blue band indicates the SGn (aa 189–451), SGn1 (aa 189–289), SGn 2 (aa 229–319), SGn 3 (aa 269–369), SGn 4 (aa 320–410) and SGn 5 (aa 349–451) segments respectively. (C) Schematic representation of epitope mapping strategy involved 16 overlapping 16mer peptides spanning SGn1 and SGn2.

peptide was constructed into prokaryotic expression vector pMAL-c2x with the Tag protein maltose binding protein (MBP) [28], in which the major steps were conducting an annealing reaction involving paired plus and minus strands, conducting a ligation reaction involving annealed DNA fragment and pMAL-c2x plasmid cut by *Bam*H I and *Sal* I, then transformed into *E. coli* (TB1 strain) [29] competent cells with the ligation mixture; screening of the r-clones by carrying out sodium dodecyl sulfate polyacrylamide gel electrophoresis (SDS-PAGE) using total proteins from each induced clone and observing whether there is a specific 16/8/10mer peptide on the gel; and sequencing of inserted DNA fragment encoding each 16/8/10mer peptide for each determined r-clone to ensure that all synthesized DNA sequences are accurate. The expression vectors were identified by DNA sequencing (Sangon biotech, Shanghai, China).

### Expression of target short peptide

Each determined r-clone was used to express a 16/8/10mer peptide in *E. coli* (TB1 strain) cells, which was fused with the MBP protein, that is, each r-clone was cultivated in 3 mL Luria Bertani (LB) medium with 100 μg/mL ampicillin at 220 rpm overnight. The following day, 30 μL of bacterial culture was added to 3 mL of fresh LB medium, grown at 37°C for 4 h to increase the bacterial density until reaching an optical density at 600 nm ($OD_{600}$) of 0.5–0.7, IPTG was added to the cultures to a final concentration of 0.2 mM, and then grown at 37°C for 4 h for inducing the expression of the target short peptide. All collected cell pellets containing the expressed 16/8/10mer peptide fusion proteins were stored at -20°C.

### SDS-PAGE and Western blot

The cell pellets obtained from 2 mL culture of expressed 16/8/10mer peptide were boiled at 95°C in 200 μL of 1×SDS-PAGE loading buffer for 10 min, and the proteins were resolved by 15% SDS-PAGE under reducing conditions. Gels were either stained with Coomassie brilliant blue G-250 for analyzing the bands corresponding to the target 16/8/10mer peptide, or used for Western blot by electro transferring the proteins onto a 0.2 μm nitrocellulose (NC) membrane [30]. Regarding the specific antigen-antibody reaction, the NC membrane was blocked with 5% (w/v) skimmed milk powder in Tris-buffered saline-Tween 20 (TBS-T), treated with rabbit pAb α-SGn (1:1000 dilution) or sheep serum (1: 100 dilution) as the primary antibody, and then reacted with goat anti-rabbit IgG or mouse anti-goat IgG conjugated to HRP (1:5000 dilution, Trans-Gen Biotech) as the secondary antibody. Finally, the blot was performed using the ECL plus Western blot detection reagent according to the manufacturer's instructions, and it was then imaged by GE-Image Quant LAS 4000 (GE Healthcare, Buckinghamshire, UK).

### Sequence alignment of homologous SFTSVs

To assess the conservation of each identified epitope among SFTSV homologous proteins, 13 SGn aa sequences from different genetic lineages were obtained from GenBank based on the phylogenetic tree of SFTSV strains [9]. The aa sequences of the SGn segments from the strain WCH/97/HN/China/2011 (GenBank code: AFH88227.1) and other 12 homologous proteins were aligned using the ClustalW program [31] and visualized using Genedoc [32].

## Results

### Antigenicity identification of the truncated Gn segments

To identify the antigenicity of the truncated Gn and the efficacy of rabbit pAb α-SGn [9], Western blot and immunofluorescence assay (IFA) were carried out. Expressed recombinant

SGn (r-SGn) was separated by SDS-PAGE (Fig 2A). The Western blot results showed that r-SGn (29 kDa) had an apparent effect on binding to rabbit pAb α-SGn (Fig 2B). The IFA was conducted to further confirm the reactivity of rabbit pAb α-SGn with the viral particle. The IFA results showed that the rabbit pAb α-SGn reactive with the SFTSV viral particle, but the

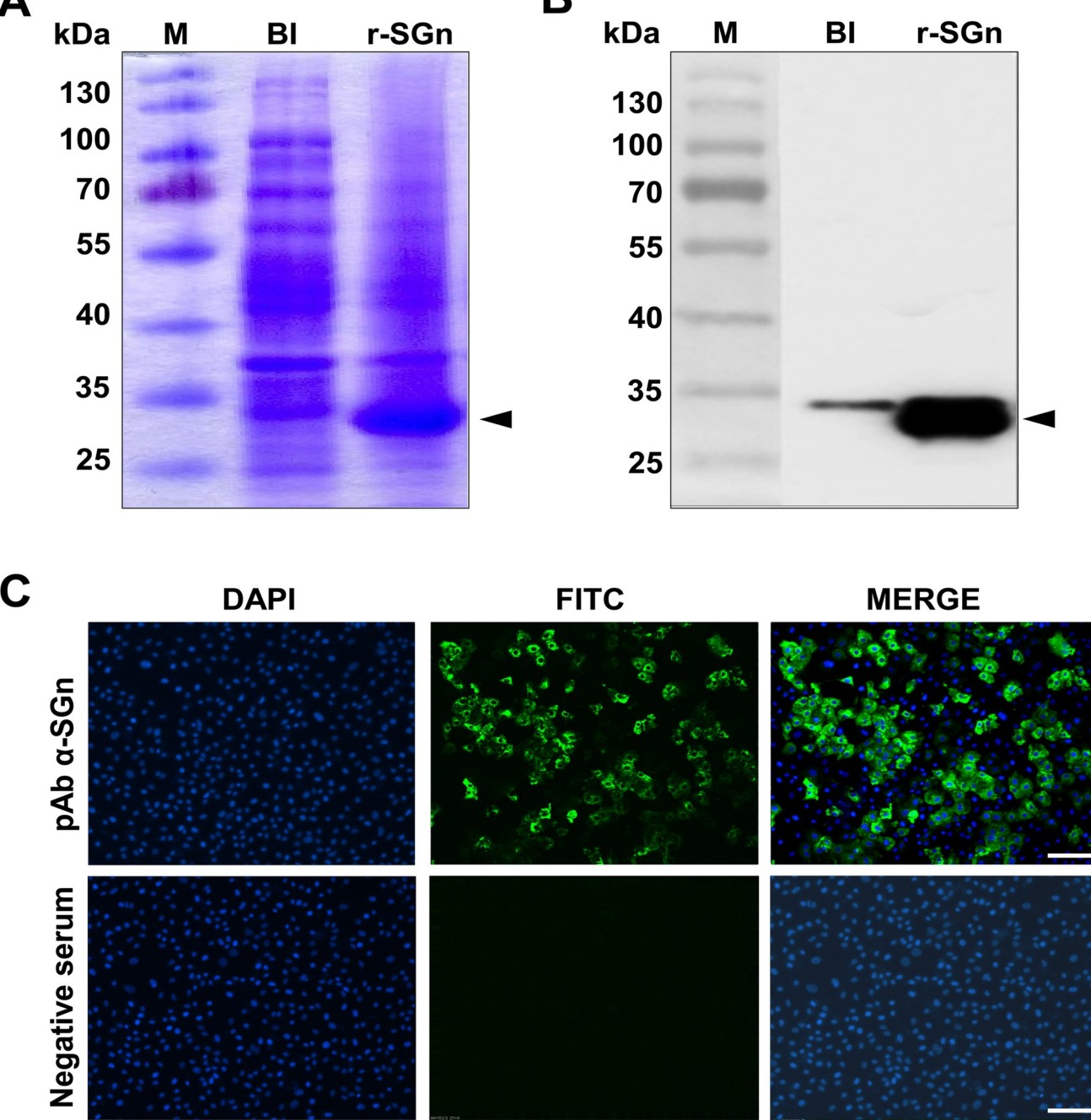

**Fig 2. SDS-PAGE and Western blot analysis of r-SGn expression and identification of reactivity of rabbit pAb α-SGn.** (A) SDS-PAGE analysis of r-SGn expression. M: standard protein marker; BI, total bacterial protein before IPTG induction. (B) Identification of the antigenicity of r-SGn by Western blot using rabbit pAb α-SGn. (C) Identification of the reactivity of pAb α-SGn with the viral particle in Vero cells by IFA; bars indicate 100 μm.

negative sera were not (Fig 2C). These Western blot and IFA results suggest that r-SGn has efficient antigenicity and pAb α-SGn has an effective ability to recognize antigens.

To further identify the antigenicity of truncated overlapping r-SGn, truncated fragments were expressed using the prokaryotic expression vector pET-32a, and each predicted protein was fused with Trx tag, S tag and His tag (with a size about 18 kDa). The SDS-PAGE result showed the expressed recombinant proteins of r-SGn 1 (27 kDa), r-SGn 2 (28 kDa), r-SGn 3 (28 kDa), r-SGn 4 (26 kDa), and r-SGn 5 (27 kDa). All recombinant proteins were detected by Western blot analysis using the rabbit α-SGn or anti-His antibody. The result showed that fusion proteins r-SGn1, r-SGn2, and r-SGn3 were reacted with rabbit pAb α-SGn, but not the other fusion expressed proteins (Fig 3). Therefore, we concluded that the region of Gn aa 189–323 contains an epitope that can be recognized by rabbit pAb α-SGn.

## Mapping epitopes on the SGn1 and SGn2 segments

To determine the existence of epitopes on the SGn1 and SGn2 segments, the immunodominant region aa 189 to aa 323 of SGn were truncated into 16 16mer peptides using BSPs (P1-P16). The 16 overlapping 16mer peptides were fusion expressed with MBP in *E. coli*, and the fusion proteins were observed as approximately 42 kDa. Among the 16 expression products, the following six 16mer peptides were identified as positive by Western blot: P1, P6, P9, P12, P13, and P16 (Fig 4A). The sensitivity of the antigen-antibody reaction was determined using the same quantity of peptides for the Western blot detection, and the data were quantitatively analyzed using Image-J software (http://rsb.info.nih.gov/ij/). The relative grayscale

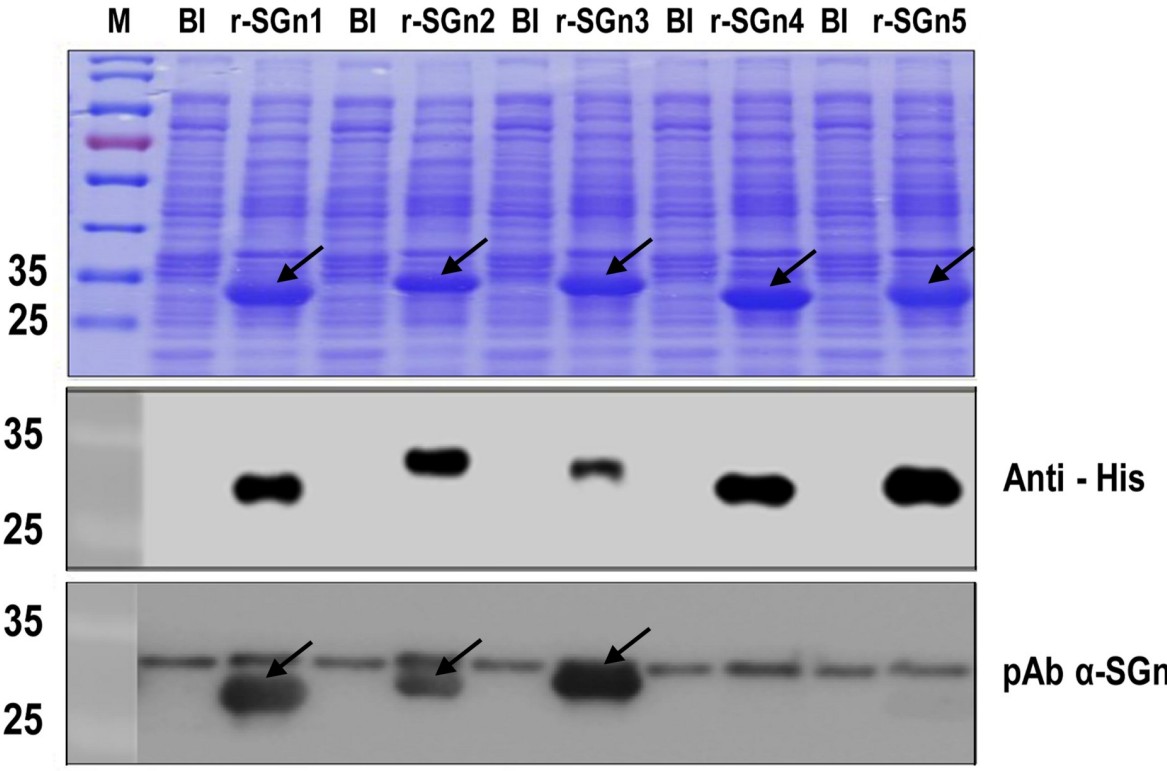

**Fig 3. Prokaryotic expression and immunoblot analysis of truncated SGn segments.** SDS-PAGE and Western blot analysis of expressed r-SGn1, r-SGn2, r-SGn3, r-SGn4, and r-SGn5 using rabbit pAb α-SGn. BI, total bacterial protein before IPTG induction. The arrows represent the three expressed target segments on the gel and the reactive segments in the Western blot analysis. Samples were loaded on the same gels and were processed in parallel.

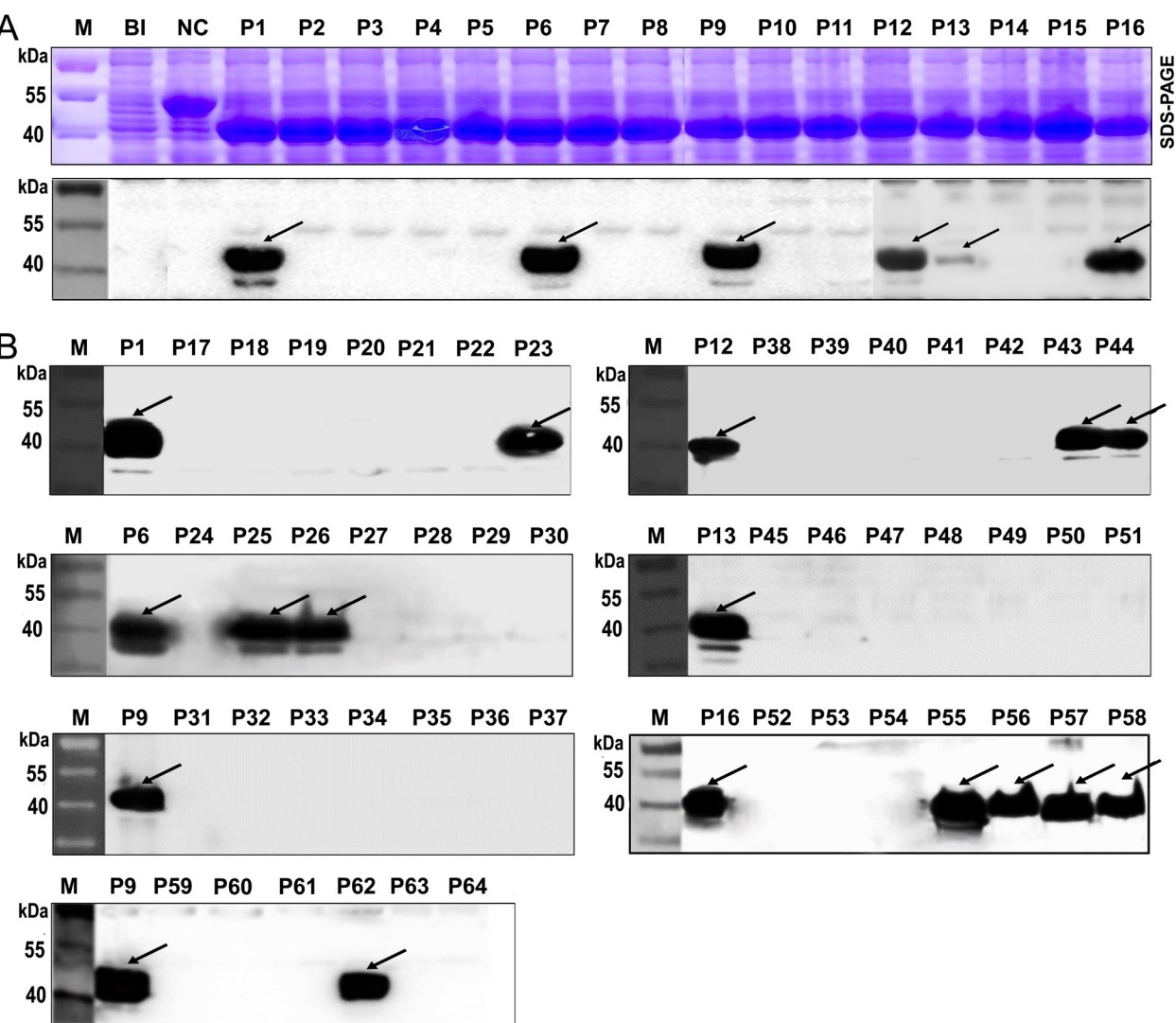

**Fig 4. SDS-PAGE and Western blot analysis of MBP fusion proteins expressed 16/8/10mer peptides derived from SGn.** (A) Western blot analysis of expressed 16mer peptides using rabbit pAb α-SGn. (B) Western blot analysis of 42 expressed 8mer peptides and 6 expressed 10mer peptides. BI, total bacterial protein before IPTG induction. NC, negative control (MBP protein expressed by pMAL-c2x). Samples were loaded on the same gels and were processed in parallel. The arrows indicate the 16/8/10mer peptides with a positive antigen-antibody reaction in the Western blot analysis.

results showed that 16mer peptides P1, P6 and P9 had a significantly (P<0.05) higher sensitivity for the antigen-antibody reaction (S1 Fig).

To refine and further mapping epitopes on SGn, the positive 16mer peptides were screened in the next round involving 8/10mer peptides. All 48 biosynthetic overlapping 8/10mer peptide aa sequences and corresponding sites on SGn are shown in S2 Table. The Western blot results showed that 8mer peptide P23 (FSQSEFPD) derived from the 16mer peptide P1, was recognized by rabbit pAb α-SGn (Fig 4B). This indicates that the minimal motifs of the epitopes within P1 was [196]FSQSEFPD[203] (designated epitope, E1), based on the shared residues (Fig 4B). Similarly, other antigenic peptides were further identified and analyzed (Figs 4B and 5). The fine epitopes were [232]GHSHKII[238] (E2) in P6, [256]VCYKEGTGPC[265] (E3) in P9, [285]FCKVAG[290] (E4) in P12 and P13, [316]SYGGM[320] (E5) in P16.

| Peptide items | Amino acids | Position in Gn | Peptide items | Amino acids | Position in Gn |
|---|---|---|---|---|---|
| P17 | FLELKSFS | 190-197 | P41 | GDMQFCKV | 281-288 |
| P18 | LELKSFSQ | 191-198 | P42 | DMQFCKVA | 282-289 |
| P19 | ELKSFSQS | 192-199 | P43 | MQFCKVAG (E4) | 283-290 |
| P20 | LKSFSQSE | 193-200 | P44 | QFCKVAGC (E4) | 284-291 |
| P21 | KSFSQSEF | 194-201 | | | |
| P22 | SFSQSEFP | 195-202 | P45 | CKVAGCEH | 286-293 |
| P23 | E1 — FSQSEFPD | 196-203 | P46 | KVAGCEHG | 287-294 |
| | | | P47 | VAGCEHGE | 288-295 |
| P24 | DVGHSHKI | 230-237 | P48 | AGCEHGEE | 289-296 |
| P25 | VGHSHKII (E2) | 231-238 | P49 | GCEHGEEA | 290-297 |
| P26 | GHSHKIIM (E2) | 232-239 | P50 | CEHGEEAS | 291-298 |
| P27 | HSHKIIMR | 233-240 | P51 | EHGEEASE | 292-299 |
| P28 | SHKIIMRE | 234-241 | | | |
| P29 | HKIIMREH | 235-242 | P52 | PGEVVVSY | 310-317 |
| P30 | KIIMREHK | 236-243 | P53 | GEVVVSYG | 311-318 |
| | | | P54 | EVVVSYGG | 312-319 |
| P31 | DFVCYKEG | 254-261 | P55 | VVVSYGGM (E5) | 313-320 |
| P32 | FVCYKEGT | 255-262 | P56 | VVSYGGMR (E5) | 314-321 |
| P33 | VCYKEGTG | 256-263 | P57 | VSYGGMRV (E5) | 315-322 |
| P34 | CYKEGTGP | 257-264 | P58 | SYGGMRVR (E5) | 316-323 |
| P35 | YKEGTGPC | 258-265 | | | |
| P36 | KEGTGPCS | 259-266 | P59 | KDFVCYKEGT | 253-262 |
| P37 | EGTGPCSE | 260-267 | P60 | DFVCYKEGTG | 254-263 |
| | | | P61 | FVCYKEGTGP | 255-264 |
| P38 | SCRGDMQF | 278-285 | P62 | E3 — VCYKEGTGPC | 256-265 |
| P39 | CRGDMQFC | 279-286 | P63 | CYKEGTGPCS | 257-266 |
| P40 | RGDMQFCK | 280-287 | P64 | YKEGTGPCSE | 258-267 |

**Fig 5. Synthetic 8/10mer peptide sequences derived from a span of the immunodominant peptides.** The yellow highlighting represents the common sequences among immunodominant peptides that react with rabbit pAb α-SGn according to Western blot analysis.

### Cross-reactivity of the identified epitope motifs with anti-SFTSV serum

To determine whether the minimal epitopes are rabbit specific or also recognizable by other host species, five randomly selected 8/10mer peptides, each of which containing one of the five identified BCEs, were subjected to Western blot using sera from sheep with or without SFTSV infection. Results showed that, P23 (containing E1), P25 (containing E2), P62 (containing E3), P44 (containing E4) and P56 (containing E5) were reacted with SFTSV positive sheep sera, while all the epitopes were not reacted with the SFTSV antibody-negative sheep sera (Fig 6).

### 3D structures of the minimal motifs of the identified epitopes and sequence conservation analysis

PyMOL[TM] software was used to simulate the 3D structure of SFTSV-Gn to locate all the mapped epitopes. The results showed that epitopes are located on the surface of the SFTSV-Gn protein (Fig 7). Epitopes E1 ([196]FSQSEFPD[203]), E2 ([232]GHSHKII[238]), and E5 ([316]SYGGM[320]) are located on domain II, E3 ([256]VCYKEGTGPC[265]) and E4 ([285]FCKVAG[290]) are located on domain III of SFTSV-Gn, all the epitopes were located on a well-defined helix-loop-helix structure (Fig 7A and 7B). The antigenic index and hydrophilicity plot for the SFTSV-Gn aa 189–321 were obtained using the methods of Jameson-Wolf [33] and Kyte-Doolittle [34], respectively. All five epitopes exhibited both high antigenicity and hydrophobicity, this is consistent with the antigenic principles of surface accessibility and hydrophilicity (Fig 7C).

To assess the conservation of each identified epitope among SFTSV homologous proteins, 13 Gn aa sequences from different countries and genetic lineages were obtained from GenBank based on the phylogenetic tree of SFTSV strains [9]. As far as we know, 13 complete Gn

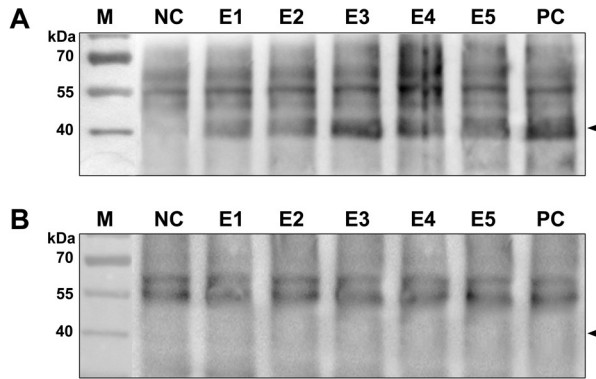

**Fig 6. Western blot of five 8/10mer peptides containing identified epitopes performed using positive sera from sheep with a confirmed history of SFTSV infection.** (A) A positive serum sample from sheep with a confirmed history of SFTSV infection. (B) A serum sample from sheep with no history of SFTSV infection was used as a negative control. NC, negative control (MBP protein expressed by pMAL-c2x). PC, positive control (16mer peptide P1 recognized by sheep positive serum).

sequences of SFTSV strains isolated in China have been registered in the GenBank database, and these were all compared to the Gn sequence investigated in this study. The aa sequences of the SGn segments from the SFTSV strain WCH/97/HN/China/2011 (GenBank code: AHH92824.1) and other SGn homologous proteins were aligned using the ClustalW program and visualized using Genedoc. The SFTSV strains selected were representative of eight genetic lineages: C1 (China, ADZ04477.1), C2 (China, AOO85597.1; South Korea, AGT98506.1), C3 (China, AHE38391.1), C4 (China, AGI97040.1), C5 (Japan, BAQ59257.1), J1 (Japan, BAQ59263.1; South Korea, APT42346.1), J2 (China, ANC60451.1; Japan, BAQ59261.1) and J3 (Japan, BAN58187.1; China, AMK05828.1). The comparison of the 13 SGn sequences indicated that epitopes E1, E2, E3, E4 and E5 were fully conserved (Fig 8). Therefore, these five epitopes can be used as candidate antigen peptides in SFTSV general diagnostic studies.

## Discussion

SFTSV is a severe emerging hemorrhagic fever virus that causes thousands of people to be hospitalized each year [2]. No vaccines or effective drugs against SFTSV have yet been developed. Understanding the life cycles of the viral infection may inform the development of important antiviral strategies. The SFTSV glycoprotein Gn/Gc mediates virus entry by binding to cellular receptors and inducing the fusion of the virus to the cell membrane during endocytosis [35]. Gn/Gc may function as an inducer that elicits the production of neutralizing antibodies [16]. Sun and colleagues showed that recombinant SFTSV-Gn bound to susceptible cell lines, Gn might act as a membrane anchoring protein during viral entry into target cells [17, 36].

Classification by protein topology, SFTSV-Gn was a type I transmembrane protein [37] that contained the N-terminal ectodomain dominated the cell surface binding capability. The SFTSV-Gn structure can be divided into the head and stem domains, the head domain corresponds to Gn 20–337, and the stem domain corresponds to Gn 338–452. The stem domain was responsible for the dimerization of SFTSV-Gn via disulfide bonds [23, 38]. However, the structure of the Gn stem domain remains unknown. Gn was also the target for neutralization antibodies, indicates that it is an important protein for prevention and vaccine design, as well as the study of viral infection mechanism of SFTSV. Recently, Wu *et al.* [23] reported the crystal structures of Gn head domains from SFTSV, and indicated that SFTSV-Gn can be reacted with mAb 4–5, a neutralizing monoclonal single chain fragment variable (ScFv) antibody

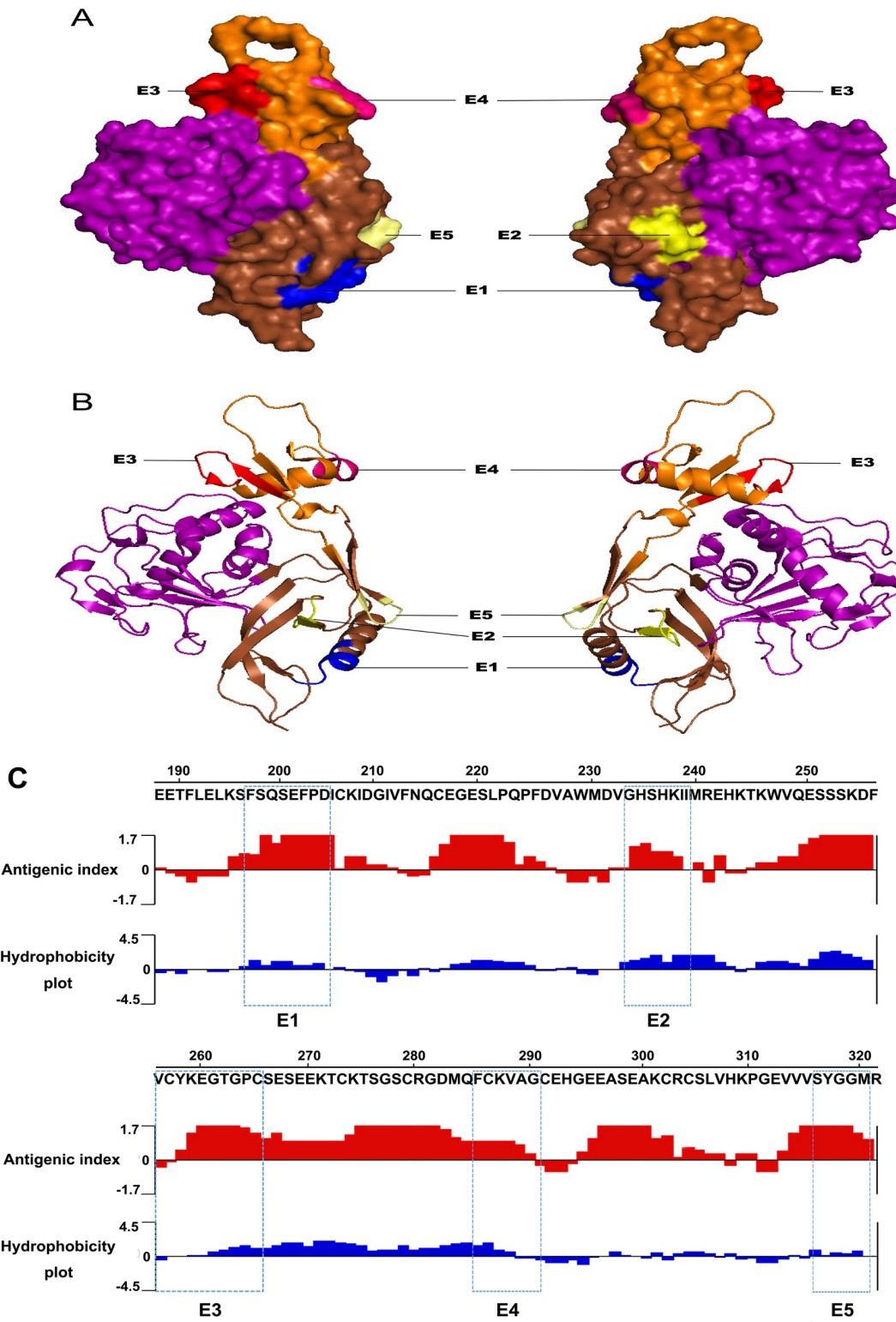

**Fig 7. Position of the minimal motifs of the mapped epitopes on the predicted 3D and secondary structure SFTSV-Gn.** The surface (A) and cartoon (B) modeling shows the overall 3D structure of SFTSV-Gn from strain WCH/97/HN/China/2011 (PDB code: 5Y11). The five minimal epitopes are located on the SFTSV-Gn domain II (brown) and III (orange), but not domain I (purple). The molecular surfaces of the five minimal epitopes are shown in different colors (E1, blue; E2, yellow; E3, red; E4, hotpink; E5, paleyellow). The figures were generated using the PyMOL™ molecular graphics system. (C) The

antigenic index and hydrophilicity plot prediction for aa residues189-321 of the Gn sequence of SFTSV using DNAStar Protean software. Epitopes were highlighted in blue rectangles.

cloned from SFTS recovered patient [22]. The structure of the Gn head domain, which was composed of three subdomains including domain I, II and III. MAb 4–5 binds to domain III of SFTSV-Gn. The neutralizing effect of mAb4-5 has been shown only *in vitro*. However, it's *in vivo* efficacy remains to be investigated. Kim *et al.* [21] reported that mAb 10 binding to Gn was predicted to be affected by domain II and the stem region of SFTSV-Gn. Unlike the mAb4-5, the mAb10 showed a potent neutralizing effect *in vitro* and protective effect *in vivo*.

In the present study, we applied the modified BSP method to identify the immunodominant BCEs in the SGn from SFTSV. The SFTSV strain WCH/97/HN/China/2011 was selected for fine epitopes mapping. Several outstanding merits of BSP/MBP method are summarized as follows: i) using the truncated MBP as a carrier, it makes the expressed 16/8/10mer peptides fusion proteins in the weak antigenic area of bacterial proteins, and thus permits using them to map the BCEs of target protein; ii) it is simple, cost-effective and cheaper compared with other epitope mapping methods [24, 25]. Five linear epitopes were identified using rabbit pAb α-SGn. The epitopes comprised are listed as E1 ($^{196}$FSQSEFPD$^{203}$), E2 ($^{232}$GHSHKII$^{238}$), and E5 ($^{316}$SYGGM$^{320}$) on domain II, E3 ($^{256}$VCYKEGTGPC$^{265}$) and E4 ($^{285}$FCKVAG$^{290}$) on domain III of SFTSV-Gn. As described above, mAb4-5 and mAb 10 bind to domain III and II, respectively, indicating that these subdomains were highly antigenic regions on the SFTSV-Gn. It is

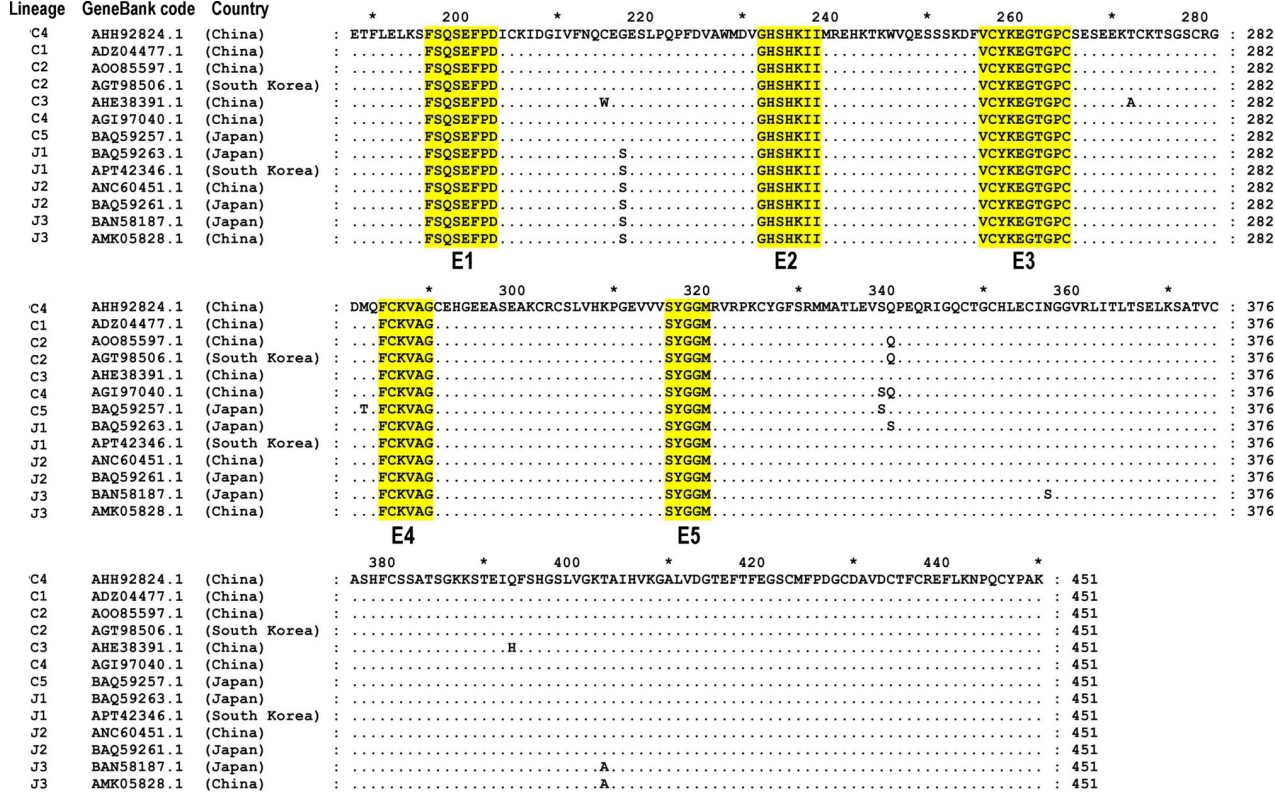

**Fig 8. Sequence alignment of the homologous SGn segments from SFTSV strains.** The GenBank codes and sources are shown on the left. The five minimal epitopes E1, E2, E3, E4 and E5 recognized by rabbit pAb α-SGn are highlighted in yellow. Dots (.) indicate identical aa residues within all 13 strains.

worth noting that, the epitopes recognized by mAb 4–5 or mAb 10 were conformational and not linear to the paratope [21, 23].

The sequences of 13 strains of SFTSV from different countries and lineages were being analyzed. The results showed that five epitopes (E1, E2, E3, E4 and E5) were fully conserved among the 13 SFTSV strains (Fig 8). Therefore, these five BCEs can be used as a selectable candidate for general diagnostic studies. The SFTSV-Gn 3D structure and the distribution of the five epitope motifs of SGn were analyzed using PyMOL^TM software. The results showed that five epitopes were exposed on the surface of the 3D structure, contained in the flexible helix-loop-helix region (Fig 7), indicating that they could easily bind to the antibodies. The epitopes located on the surface of the target protein play an important role in the future development of drugs that interact with target antigens. It is generally believed that the multi-epitope based recombinant vaccines offer numerous advantages compared to the protein-based vaccine. These advantages are including multi-valency cost- effectiveness, economical production and stability under different ambient conditions. Moreover, the efficacy of multi-epitope vaccines can be further improved by combining helper T cells and promiscuous epitopes, and adopting toll-like receptor ligands as an adjuvant [39, 40]. However, we have not yet known that the identified five BCEs of SGn in this study will obtain the neutralizing activity. Further study in this direction is needed for verification.

In conclusion, the five high conserved linear BCEs (E1, E2, E3, E4 and E5) were recognized with rabbit pAb α-SGn. All the five epitopes interacted with the sheep serum infected naturally with SFTSV. Our results will escalate the understanding of the epitope distribution and function of SFTSV-Gn, and provide fundamental information for the elucidation of the design and development of a SFTSV multi-epitope peptide detection antigen.

## Supporting information

**S1 Table. 16mer peptides amino acid sequence and its location on SFTSV-Gn.**
(DOC)

**S2 Table. 8/10mer peptides amino acid sequence and its location on SFTSV-Gn.**
(DOC)

**S1 Fig. Relative grayscale level analyses of 16/8/10mer peptides from Western blot using Image-J software.** To determine the sensitivity of the antigen-antibody reaction involving the 16mer (A) and 8/10mer (B) peptides, quantitative analyses were performed using the same quantity of peptides for detection. The relative grayscale level of each 16/8/10mer peptide compared to the positive 16mer peptide P1 was analyzed according to the results in Fig 3. Statistical analysis of data was performed using one-way analysis of variance (ANOVA) to determine the significant differences using SPSS software. Letters (a, b, c) indicate the significant differences ($P<0.05$).
(TIF)

**S1 Raw images.**
(PDF)

## Author Contributions

**Funding acquisition:** Fei Deng, Surong Sun.

**Investigation:** Abulimiti Moming.

**Methodology:** Abulimiti Moming, Shu Shen, Bo Wang, Juntao Ding, Surong Sun.

**Resources:** Shen Shi, Shu Shen, Jie Qiao, Xihong Yue, Yujiang Zhang.

**Software:** Abulimiti Moming, Bo Wang.

**Supervision:** Zhihong Hu, Fei Deng, Yujiang Zhang, Surong Sun.

**Writing – original draft:** Abulimiti Moming.

**Writing – review & editing:** Surong Sun.

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
