## [Decision Letter · Decision Letter 0]

23 Dec 2020

PONE-D-20-32530

Fine mapping epitope on Glycoprotein-Gn from Severe Fever with Thrombocytopenia Syndrome Virus

PLOS ONE

Dear Dr. Sun,

Thank you for submitting your manuscript to PLOS ONE. After careful consideration, we feel that it has merit but does not fully meet PLOS ONE’s publication criteria as it currently stands. Therefore, we invite you to submit a revised version of the manuscript that addresses the points raised during the review process.

1) Please provide more details of the antiserum used to map the Severe Fever with Thrombocytopenia Syndrome Vírus Glycoprotein. How was it generated? What is it specificity? Was is purified? How?

2) The authors did not show that the linear epitopes are protective or induce neutralizing antibodies against the vírus. Modify the discussion accordingly;

3) Please, see the comments raised by both the reviewers.

We look forward to receiving your revised manuscript.

Kind regards,

Paulo Lee Ho, Ph.D.

Academic Editor

PLOS ONE

Journal Requirements:

2. In your Methods section, please specify the source of the serum samples from the healthy rabbit and healthy sheep used for your study.

3.We suggest you thoroughly copyedit your manuscript for language usage, spelling, and grammar. If you do not know anyone who can help you do this, you may wish to consider employing a professional scientific editing service.  

Reviewers' comments:

Reviewer's Responses to Questions

**Comments to the Author**

1. Is the manuscript technically sound, and do the data support the conclusions?

Reviewer #1: Yes

Reviewer #2: Yes

2. Has the statistical analysis been performed appropriately and rigorously? 

Reviewer #1: I Don't Know

Reviewer #2: N/A

3. Have the authors made all data underlying the findings in their manuscript fully available?

Reviewer #1: Yes

Reviewer #2: Yes

4. Is the manuscript presented in an intelligible fashion and written in standard English?

Reviewer #1: Yes

Reviewer #2: Yes

5. Review Comments to the Author

Reviewer #1: Moming A et al. have mapped the epitopes on glycoprotein Gn of SFTSV. The study is interesting and the work was done very carefully and very well. There are few mistakes need to be changed in the manuscript.

1. Throughout the manuscript Western should be capitalized for Western blot.

2. The following method description in the result section should be deleted.

1) Between lines 221 and 224, “Expressed recombinant SGn (r-SGn) was first separated by SDS-PAGE (Fig 2A) and then transferred to nitrocellulose membranes; pAb α-SGn (1:1000 dilution) was used as the primary antibody, and HRP-conjugated goat anti-rabbit IgG (1:5000 dilution) was used as the secondary antibody”.

2) Between lines 257 and 259, “NC membranes were blocked with 5% (w/v) skimmed milk powder, incubated sequentially with rabbit pAb α-SGn (1:1000 dilution), and incubated with goat anti-rabbit (1:5000 dilution), and then visualized by ECL”.

3) Between lines 277 and 279, “NC membranes were blocked with 5% (w/v) skimmed milk powder, incubated sequentially with rabbit pAb α-SGn (1:1000 dilution) and goat anti-rabbit IgG (1:5000 dilution), and then visualized by ECL”.

3. Missing references:

1) Online 59, a reference (Yu XJ, Liang MF, Zhang SY, et al. Fever with thrombocytopenia associated with a novel bunyavirus in China. N Engl J Med. 2011;364(16):1523-1532) should be added.

2) On line 65, SFTSV is not Banyangvirus. You need change the name of the virus to Bandavirus and use the reference “Kuhn JH, Adkins S, Alioto D, et al. 2020 taxonomic update for phylum Negarnaviricota (Riboviria: Orthornavirae), including the large orders Bunyavirales and Mononegavirales. Arch Virol. 2020;165:3023-3072” to replace reference 12.

3) Online 65, you need add references for the statement “Studies have indicated that GP mediate the first step in the virus replication cycle viral entry, and are the only targets for neutralizing antibodies”

Reviewer #2: The manuscript entitles “Fine mapping epitope on Glycoprotein-Gn from Severe Fever with Thrombocytopenia Syndrome Virus” describes the linear epitope mapping of Gn protein of SFTS virus. The authors showed the five linear epitopes in the Sn protein, 196-203, 232-238, 256-265, 285-290, and 316-320 amino acid residues, respectively. The epitopes were mapped using rabbit anti Sgn antibody and the epitopes were also recognized by SFTS antibody-positive sheep sera. The manuscript is well written and the methods to map the epitopes were appropriate. However, there are two major criticism as written in the “Major points”.

Major points:

The epitopes were mapped using rabbit pAb a-SGn (reference No8) supplied from Prof. Deng, however in the reference manuscript any detail of the rabbit a-SGn was written. Thus, the authors should describe the detail of the antibody as to how the rabbit serum was prepared, whether the antibody has viral neutralizing antibody against SFTSV.

From the data shown in the manuscript, it is too much to discuss “these linear epitopes identified in the manuscript provide fundamental data for the elucidation of the design and development of a SFTSV multi-epitope peptide vaccine”, since the authors never showed the antibodies to these linear epitopes have viral neutralizing capacity of not. Actual;ly, the authors describe “However, whether the identified five BCEs of SGn in this study have neutralizing activity require further verification (line 395-396). The reviewer agrees that the data shown in the manuscript provide useful data for the development of a novel SFTS antibody detection antigen. Thus, the authors are recommended to remove the sentence describing vaccine development.

Minor points;

Line 62: “including goats, cattle, dogs, and chickens [8-10]” lacks cats as susceptible animal. Scientific Reports, 19(1): 11990 (2019) showed that cats were susceptible for SFTSV and the virus caused lethal infection in cats. The authors should add this manuscript to the reference.

Line 110: “involving” should be “involve”

Line 142-143: “computer controlled microscope” should be changed to “ the name of microscope product name”

Line 143: “Olympics” is “Olympus” ??

Line 269: “ … MBP fusion expressed …” should be “… MBP fusion proteins expressed …

Line 364 -365: “mAb 4-5, a neutralizing antibody identified in SFTS recovered patients” is not precise. mAb 4-5, a neutralizing monoclonal ScFv antibody cloned from SFTS recovered patient.”

6. PLOS authors have the option to publish the peer review history of their article (what does this mean?). If published, this will include your full peer review and any attached files.

Reviewer #1: No

Reviewer #2: No

---

## [Author Response · Author response to Decision Letter 0]

12 Jan 2021

Reviewer #1: Moming A et al. have mapped the epitopes on glycoprotein Gn of SFTSV. The study is interesting and the work was done very carefully and very well. There are few mistakes need to be changed in the manuscript.

1. Throughout the manuscript Western should be capitalized for Western blot. 

Response: Thanks for the good remind. We have revised Western to capitalized Western blot in the manuscript.

2. The following method description in the result section should be deleted.

1) Between lines 221 and 224, “Expressed recombinant SGn (r-SGn) was first separated by SDS-PAGE (Fig 2A) and then transferred to nitrocellulose membranes; pAb α-SGn (1:1000 dilution) was used as the primary antibody, and HRP-conjugated goat anti-rabbit IgG (1:5000 dilution) was used as the secondary antibody”.

2) Between lines 257 and 259, “NC membranes were blocked with 5% (w/v) skimmed milk powder, incubated sequentially with rabbit pAb α-SGn (1:1000 dilution), and incubated with goat anti-rabbit (1:5000 dilution), and then visualized by ECL”.

3) Between lines 277 and 279, “NC membranes were blocked with 5% (w/v) skimmed milk powder, incubated sequentially with rabbit pAb α-SGn (1:1000 dilution) and goat anti-rabbit IgG (1:5000 dilution), and then visualized by ECL”.

Response: Thanks for the kind suggestion. We have deleted the above method description in the result section in the revised manuscript.

3. Missing references:

1) Online 59, a reference (Yu XJ, Liang MF, Zhang SY, et al. Fever with thrombocytopenia associated with a novel bunyavirus in China. N Engl J Med. 2011;364(16):1523-1532) should be added.

Response: Thanks for the valuable suggestion. We have added this reference in line 37.

. 

2) On line 65, SFTSV is not Banyangvirus. You need change the name of the virus to Bandavirus and use the reference “Kuhn JH, Adkins S, Alioto D, et al. 2020 taxonomic update for phylum Negarnaviricota (Riboviria: Orthornavirae), including the large orders Bunyavirales and Mononegavirales. Arch Virol. 2020;165:3023-3072” to replace reference 12.

Response: Thanks for the good remind. We apologize for this oversight, the classification of SFTSV was updated according to reference (14) as Bandavirus genus within the family Phenuiviridae. Reference [12] has been replaced as [14] in the revised version in line 44.

[14] Kuhn JH, Adkins S, Alioto D, Alkhovsky SV, Amarasinghe GK, Anthony SJ, et al. 2020 taxonomic update for phylum Negarnaviricota (Riboviria: Orthornavirae), including the large orders Bunyavirales and Mononegavirales. Arch Virol. 2020 Dec;165 (12): 3023-3072. https://doi: 10.1007/s00705-020-04731-2 PMID: 32888050

3) On line 65, you need add references for the statement “Studies have indicated that GP mediate the first step in the virus replication cycle viral entry, and are the only targets for neutralizing antibodies”.

Response: Thanks for the good remind. We apologize for our earlier lack of clarity. We have added references [15] and [16] in line 48 of the revised manuscript.

[15] Tani H, Shimojima M, Fukushi S, Yoshikawa T, Fukuma A, Taniguchi S, et al. Characterization of glycoprotein-mediated entry of severe fever with thrombocytopenia syndrome virus. J Virol. 2016 May 12; 90(11): 5292-5301. https://doi: 10.1128/JVI.00110-16 PMID: 26984731

[16] Hofmann H, Li X, Zhang X, Liu W, Kühl A, Kaup F, et al. Severe fever with thrombocytopenia virus glycoproteins are targeted by neutralizing antibodies and can use DC-SIGN as a receptor for pH-dependent entry into human and animal cell lines. J Virol. 2013 Apr; 87(8): 4384-4394. https://doi: 10.1128/JVI.02628-12 PMID: 23388721

Reviewer #2: The manuscript entitles “Fine mapping epitope on Glycoprotein-Gn from Severe Fever with Thrombocytopenia Syndrome Virus” describes the linear epitope mapping of Gn protein of SFTS virus. The authors showed the five linear epitopes in the Sn protein, 196-203, 232-238, 256-265, 285-290, and 316-320 amino acid residues, respectively. The epitopes were mapped using rabbit anti Sgn antibody and the epitopes were also recognized by SFTS antibody-positive sheep sera. The manuscript is well written and the methods to map the epitopes were appropriate. However, there are two major criticism as written in the “Major points”.

Major points:

The epitopes were mapped using rabbit pAb a-SGn (reference No8) supplied from Prof. Deng, however in the reference manuscript any detail of the rabbit a-SGn was written. Thus, the authors should describe the detail of the antibody as to how the rabbit serum was prepared, whether the antibody has viral neutralizing antibody against SFTSV.

Response: Thanks for the good remind. We have provided information on antiserum used to map the Severe Fever with Thrombocytopenia Syndrome Vírus Glycoprotein Gn in the materials and methods section in line 94-102. The main preparation methods are as follows：The coding region (aa, 189-451) of SFTSV-Gn from strain WCH/97/HN/China/2011 was amplified by PCR using 2×Rapid Taq Master Mix (Vazyme Biotech, Nanjing, China) according to the manufacture’s instruction. The PCR products were cloned into the plasmid pET-28a to generate the expression plasmid pET-28a-SGn and the insert was confirmed by sequencing. Protein expression and purification were conducted as described [27]. New Zealand rabbits were injected intramuscularly with 0.5 mg of purified SGn segment and immunized at two-week intervals according to the conventional animal immune method. After the third immunization for two weeks, rabbit antiserum was separated and stored at -80℃ until use. The neutralizing activity of rabbit pAb α-SGn against SFTSV was detected using IFA, but no neutralizing effect was found.

[27] Moming A, Zhang YJ, Chang CC, Yu H, Wang MF, Hu Z, et al. Antigenicity of severe fever with thrombocytopenia syndrome virus nucleocapsid protein and its potential application in the virus serodiagnosis. Virol Sin. 2017 Feb; 32(1): 97-100. https://doi: 10.1007/s12250-016-3928-9 PMID: 28120219

From the data shown in the manuscript, it is too much to discuss “these linear epitopes identified in the manuscript provide fundamental data for the elucidation of the design and development of a SFTSV multi-epitope peptide vaccine”, since the authors never showed the antibodies to these linear epitopes have viral neutralizing capacity of not. Actually, the authors describe “However, whether the identified five BCEs of SGn in this study have neutralizing activity require further verification (line 395-396). The reviewer agrees that the data shown in the manuscript provide useful data for the development of a novel SFTS antibody detection antigen. Thus, the authors are recommended to remove the sentence describing vaccine development.

Response: Thanks for the good suggestion. We have removed the sentence describing vaccine development in the revised manuscript. 

Minor points;

Line 62: “including goats, cattle, dogs, and chickens [8-10]” lacks cats as susceptible animal. Scientific Reports, 19(1): 11990 (2019) showed that cats were susceptible for SFTSV and the virus caused lethal infection in cats. The authors should add this manuscript to the reference.

Response: Thanks for the kind remind and suggestion. We have added cats as susceptible animal for SFTSV, and added as reference [12] in line 40 in the revised manuscript.

[12] Park ES, Shimojima M, Nagata N, Ami Y, Yoshikawa T, Iwata-Yoshikawa N, et al. Severe fever with thrombocytopenia syndrome Phlebovirus causes lethal viral hemorrhagic fever in cats. Sci Rep. 2019 Aug 19; 9(1): 11990. https://doi: 10.1038/s41598-019-48317-8 PMID: 31427690

Line 110: “involving” should be “involve”

Response: Thanks for the good remind. “involving” was revised as “involve” in line 90.

Line 142-143: “computer controlled microscope” should be changed to “ the name of microscope product name”

Response: Thanks for the kind remind. “computer controlled microscope” was changed as “IX73 microscope” in lines 129.

Line 143: “Olympics” is “Olympus” ??

Response: Thanks for the good remind. “Olympics” was revised as “Olympus” in line 129.

Line 269: “ … MBP fusion expressed …” should be “… MBP fusion proteins expressed …

Response: Thanks for the good remind. “MBP fusion expressed” was revised as “MBP fusion proteins expressed” in line 250.

Line 364 -365: “mAb 4-5, a neutralizing antibody identified in SFTS recovered patients” is not precise. mAb 4-5, a neutralizing monoclonal ScFv antibody cloned from SFTS recovered patient.”

Response: Thanks for the good remind. “mAb 4-5, a neutralizing antibody identified in SFTS recovered patients” was revised as “mAb 4-5, a neutralizing monoclonal single chain fragment variable (ScFv) antibody cloned from SFTS recovered patient” in lines 342-343.

---

## [Decision Letter · Decision Letter 1]

9 Feb 2021

PONE-D-20-32530R1

Fine mapping epitope on Glycoprotein-Gn from Severe Fever with Thrombocytopenia Syndrome Virus

PLOS ONE

Dear Dr. Sun,

Thank you for submitting your manuscript to PLOS ONE. After careful consideration, we feel that it has merit but does not fully meet PLOS ONE’s publication criteria as it currently stands. Therefore, we invite you to submit a revised version of the manuscript that addresses the points raised during the review process.

We look forward to receiving your revised manuscript.

Kind regards,

Paulo Lee Ho, Ph.D.

Academic Editor

PLOS ONE

Reviewers' comments:

Reviewer's Responses to Questions

**Comments to the Author**

1. If the authors have adequately addressed your comments raised in a previous round of review and you feel that this manuscript is now acceptable for publication, you may indicate that here to bypass the “Comments to the Author” section, enter your conflict of interest statement in the “Confidential to Editor” section, and submit your "Accept" recommendation.

Reviewer #1: All comments have been addressed

Reviewer #2: All comments have been addressed

2. Is the manuscript technically sound, and do the data support the conclusions?

Reviewer #1: Yes

Reviewer #2: Yes

3. Has the statistical analysis been performed appropriately and rigorously? 

Reviewer #1: N/A

Reviewer #2: N/A

4. Have the authors made all data underlying the findings in their manuscript fully available?

Reviewer #1: Yes

Reviewer #2: Yes

5. Is the manuscript presented in an intelligible fashion and written in standard English?

Reviewer #1: Yes

Reviewer #2: Yes

6. Review Comments to the Author

Reviewer #1: The minor comments are on line 178 bacteria culture should be bacterial culture and on line 242 E. coli should be italicized.

Reviewer #2: In the revised manuscript, all the comments have been properly addressed, thus the manuscript is considered to be accepted.

7. PLOS authors have the option to publish the peer review history of their article (what does this mean?). If published, this will include your full peer review and any attached files.

Reviewer #1: **Yes: **Xue-jie Yu

Reviewer #2: No

---

## [Author Response · Author response to Decision Letter 1]

10 Feb 2021

Reviewer #1: The minor comments are on line 178 bacteria culture should be bacterial culture and on line 242 E. coli should be italicized.

Response: Thanks for the good remind. We apologize for our carelessness, we have revised “bacteria culture” as “bacterial culture” on line 178, and italicized E. coli on line 242 in the manuscript.

---

## [Editor Report · Decision Letter 2]

18 Feb 2021

Fine mapping epitope on Glycoprotein-Gn from Severe Fever with Thrombocytopenia Syndrome Virus

PONE-D-20-32530R2

Dear Dr. Sun,

We’re pleased to inform you that your manuscript has been judged scientifically suitable for publication and will be formally accepted for publication once it meets all outstanding technical requirements.

Kind regards,

Paulo Lee Ho, Ph.D.

Academic Editor

PLOS ONE
---

## [Editor Report · Acceptance letter]

22 Feb 2021

PONE-D-20-32530R2 

Fine mapping epitope on Glycoprotein-Gn from Severe Fever with Thrombocytopenia Syndrome Virus 

Dear Dr. Sun:

I'm pleased to inform you that your manuscript has been deemed suitable for publication in PLOS ONE. Congratulations! Your manuscript is now with our production department. 

Kind regards, 

on behalf of

Dr. Paulo Lee Ho 

Academic Editor

PLOS ONE